# The regulon of *Brucella abortus* two-component system BvrR/BvrS reveals the coordination of metabolic pathways required for intracellular life

Olga Rivas-Solano[1], Mathilde Van der Henst[2], Amanda Castillo-Zeledón[3], Marcela Suárez-Esquivel[3], Lohendy Muñoz-Vargas[3], Zeuz Capitan-Barrios[3¤], Nicholas R. Thomson[4], Esteban Chaves-Olarte[5], Edgardo Moreno[3], Xavier De Bolle[2], Caterina Guzmán-Verri[3]*

**1** Centro de Investigación en Biotecnología, Instituto Tecnológico de Costa Rica, Cartago, Costa Rica, **2** Unité de Recherche en Biologie des Microorganismes, Université de Namur, Namur, Belgium, **3** Programa de Investigación en Enfermedades Tropicales, Escuela de Medicina Veterinaria, Universidad Nacional, Heredia, Costa Rica, **4** Parasites and Microbes from Pathogen Genomics, Wellcome Trust Sanger Institute, Hinxton, United Kingdom, **5** Centro de Investigación en Enfermedades Tropicales, Universidad de Costa Rica, San José, Costa Rica

¤ Current address: Departamento de Microbiología y Parasitología, Facultad de Ciencias Naturales, Exactas y Tecnología, Universidad de Panamá, Ciudad de Panamá, Panamá
* catguz@una.cr

## Abstract

*Brucella abortus* is a facultative intracellular pathogen causing a severe zoonotic disease worldwide. The two-component regulatory system (TCS) BvrR/BvrS of *B. abortus* is conserved in members of the *Alphaproteobacteria* class. It is related to the expression of genes required for host interaction and intracellular survival. Here we report that *bvrR* and *bvrS* are part of an operon composed of 16 genes encoding functions related to nitrogen metabolism, DNA repair and recombination, cell cycle arrest, and stress response. Synteny of this genomic region within close *Alphaproteobacteria* members suggests a conserved role in coordinating the expression of carbon and nitrogen metabolic pathways. In addition, we performed a ChIP-Seq analysis after exposure of bacteria to conditions that mimic the intracellular environment. Genes encoding enzymes at metabolic crossroads of the pentose phosphate shunt, gluconeogenesis, cell envelope homeostasis, nucleotide synthesis, cell division, and virulence are BvrR/BvrS direct targets. A 14 bp DNA BvrR binding motif was found and investigated in selected gene targets such as *virB1*, *bvrR*, *pckA*, *omp25*, and *tamA*. Understanding gene expression regulation is essential to elucidate how *Brucella* orchestrates a physiological response leading to a furtive pathogenic strategy.

## Introduction

*Brucella* spp. are Gram-negative, facultative intracellular *Alphaproteobacteria* related to endosymbionts, animal, and plant pathogens [1]. These organisms cause brucellosis, a worldwide

**Data Availability Statement:** All relevant data are within the paper and its Supporting information files. ChIP-Seq data have been deposited in the ArrayExpress database at http://www.ebi.ac.uk/arrayexpress/experiments/E-MTAB-9740. Supplementary information has been deposited at 10.6084/m9.figshare.19450661.

**Funding:** Fondos del Sistema FEES/CONARE [02-2020 to C.G-V], Fondos FIDA, Universidad Nacional [SIA 0047-17 to C. G-V], Espacio Universitario de Estudios Avanzados, UCREA [B8762 and C0456 to E.C-O] from the presidency of University of Costa Rica, the Vice Presidency for Research, University of Costa Rica. Instituto Tecnológico de Costa Rica (ITCR) [15-15-D to O.R-S], PINN-MICITT [PND-137-15-1 to O.R-S]. Fonds de la Recherche Scientifique-Fonds National de la Recherche Scientifique (F.R.S.-FNRS) [PDR Brucell-cycle T.0060.15 to X.D.B., FRIA Doctoral Grant to M.V.d. H.], and Wellcome Trust Sanger Institute [098051 to N.R.T]. The funders had no role in study design, data collection and analysis, decision to publish, or preparation of the manuscript.

**Competing interests:** The authors have declared that no competing interests exist.

distributed and neglected zoonotic disease [2]. *Brucella abortus* induces abortion and infertility in cattle. Humans are accidental hosts, showing an acute febrile illness that may persist and evolve into a long-lasting infection with severe complications [2]. *Brucella* pathogenicity resides in its ability to invade, survive, and replicate inside host cells, including professional and non-professional phagocytes [3]. Once inside the cell, the bacterium is found within a membrane-bound compartment avoiding the lysosomal route and redirecting its trafficking to a compartment derived from the endoplasmic reticulum (ER), where it replicates [3]. Eventually, this compartment acquires autophagosome features required for cell egress and spreading [4]. An exhaustive proteomic analysis of bacteria at different time points during macrophage trafficking revealed metabolic adjustments consistent with the different conditions found in the intracellular compartments [5]. Early in infection *B. abortus* 2308 Wisconsin downregulates carbohydrate-based carbon utilization, periplasmic transporters, and protein synthesis. Alternative energy sources based mainly on anaplerotic routes and generation of glutamate by enzymatic conversion of amino acids and low oxygen tension-type of respiration are evident. At the same time, bacteria change their membrane composition and restrict their protein and nucleic acid synthesis, probably reflecting the stress conditions in the shelter vacuole [5]. The type IV secretion system VirB and apoptosis inhibitory mechanisms are critical for the survival within these *Brucella*-containing vacuoles, which in time, associate with compartments of the ER through LPS modifications, beta-cyclic glucans, and VirB effectors [3]. Twenty-four hours after infection, after reaching the ER, bacterial metabolism shows signs of complete adaptation to a low oxygen tension-type of respiration, an increase of transporters involved in the capture of amino acids, peptides and iron. Protein and nucleic acid synthesis resume, and the outer membrane´s topology shows signs of changes again according to the new environment. After two days of intracellular life, bacteria extensively replicate in a vacuole associated with the ER, restoring most of the differentially expressed proteins to pre-infection levels [5]. Bacteria then reach an autophagosome-like exit compartment where they are ready to egress from the host cell and start a new infection cycle [4].

Transitioning from an extracellular to an intracellular milieu requires a highly coordinated gene expression achieved through several regulatory mechanisms, including two-component regulatory systems (TCSs): signal transduction systems that allow bacteria to sense and respond to environmental variations [6]. The simplest TCS includes a sensor histidine kinase and a response regulator. When the histidine kinase senses external signals, it autophosphorylates on a conserved histidine residue. Then it transfers the phosphoryl group to a conserved aspartate residue in the response regulator. The phosphorylated form of this protein shows an increased affinity for DNA binding sites, activating or repressing a particular set of genes, which constitute a direct regulon [6].

*Alphaproteobacteria* closely associated with eukaryotic cells, such as *Bartonella*, *Rhizobium*, *Sinorhizobium*, *Agrobacterium*, and occasionally *Ochrobactrum*, have ortholog TCS regulating functions involving host-microbe interactions [7, 8]. In *Brucella*, this TCS is BvrR/BvrS. *B. abortus* mutants in *bvrR* and in *bvrS* are avirulent, displaying reduced invasiveness and replication failure in cells and mice [9]. Previous studies showed transient activation of BvrR through phosphorylation in bacteria grown in nutrient-rich media at neutral pH (rich conditions) [10]. Brief exposition to an acidic nutrient-limited media (stress conditions) mimics *B. abortus* intracellular environment and also induces BvrR phosphorylation [10]. Proteomics and transcriptomics studies showed that *B. abortus* BvrR/BvrS is a master regulator of cell envelope homeostasis, carbon and nitrogen metabolism, and virulence-related proteins [11–16]. Accordingly, it regulates the expression of genes coding for outer membrane proteins, such as Omp25 and Omp22, and genes involved in lipid A acylation [12, 13]. In addition, a *bvrR*-

deficient *B. abortus* strain expresses reduced levels of TamB [15], a protein involved with TamA in cell envelope biogenesis, cell division, virulence, and intracellular growth [17, 18].

BvrS likely senses changes in pH and nutrients, such as the acidic and low nutrient environment found during the endosomal route. Activated BvrS triggers a transcriptional response that includes activation of a virulence circuit composed of the phosphorylated cognate regulator BvrR (BvrR-P), the quorum-sensing regulator VjbR and the type IV secretion system VirB (T4SS VirB) required to redirect bacterial trafficking to the ER [19]. Later in infection, BvrR/BvrS also senses environmental cues that reactivate the virulence circuit required to exit the host cell and increase bacterial infectivity [19].

Conserved synteny is observed in Rhizobiales genomic regions encoding ortholog TCS and downstream genes, encoding a nitrogen-related phosphotransferase system (PTS$^{Ntr}$). The PTS$^{Ntr}$ is a global regulatory mechanism used to reach metabolic fitness according to carbon and nitrogen availability by *Rhizobium leguminosarum* [20] and probably by *Brucella melitensis* [21] and *Sinorhizobium meliloti* [22]. A coordinating role between this TCS and PTS$^{Ntr}$ could allow bacteria to regulate the metabolic crossroad between carbon and nitrogen sources and adjust to the environments encountered during host cell interaction [23]. Recent evidence suggests that in *Rhizobium leguminosarum* this coordination is exerted by direct interaction between PtsN1 and the response regulator ChvI [20].

Here we confirm the relationship between BvrR/BvrS and the Pts$^{Ntr}$ system, showing that *B. abortus bvrR* and *bvrS* belong to an operon of 16 genes with conserved synteny in analyzed *Alphaproteobacteria* genomes. Furthermore, we expand our knowledge of the BvrR/BvrS regulon, describing genomic regions bound directly by BvrR-P under conditions that mimic the intracellular environment confronted by *B. abortus* while trafficking to its replicative niche. Some of these regions were related to genes encoding enzymes at the metabolic crossroads of carbon and nitrogen pathways, reinforcing the role that BvrR/BvrS has in the coordination of gene expression required for a successful *B. abortus* infection.

## Methods

### Bacterial strains and growth conditions

*B. abortus* 2308 Wisconsin (2308W) was used as a wild-type strain [24], and its derivative *B. abortus* 65.21 *bvrR*::Tn5, a BvrR-deficient strain, was used as control [9]. Both strains were grown at 37˚C in Tryptic Soy Broth (TSB), pH 7.2. All procedures involving live *B. abortus* were performed following the "Reglamento de Bioseguridad de la CCSS 39975–0", 2012, after the "Decreto Ejecutivo #30965-S", 2002 and research protocol SIA 0652–19 approved by the National University, Costa Rica.

### RNA extraction, RT-PCR, and conservation analysis of *bvrR/bvrS* operon

For co-transcriptional analysis of *bvrR* and downstream genes, total RNA isolation and RT-PCR were performed as previously described [11] from *B. abortus* 2308W cultures grown in TSB at the log and stationary growth stages. S1 Table lists the primers used for this purpose. The PCR products were analyzed on agarose gels using standard procedures [25]. The obtained amplicons were sequenced using the Big Dye terminator kit 3.1 (Life Technologies), following manufacturer instructions.

The conservation analysis of the *bvrR/bvrS* operon homologs in representative *Alphaproteobacteria* was performed using BLAST. *B. suis* 1330 genome was used as a reference for this comparison since it has been re-sequenced and is one of the largest *Brucella* genomes.

Artemis [26] and Artemis Comparison Tool were used to visualize the results [27]. The 16s rRNA genes were used for molecular phylogenetic reconstruction by the Maximum Likelihood

method based on the Tamura-Nei model [28] to infer the evolutionary history of the *Alphaproteobacteria* selected genomes. The bootstrap consensus tree inferred from 500 replicates [29] was taken to represent the evolutionary history of the taxa analyzed. The analysis involved 17 nucleotide sequences (S2 Table). All positions containing gaps and missing data were eliminated. There were a total of 1191 positions in the final dataset. Consistent with previous reports [30], an outgroup conformed by *Escherichia coli*, *Ralstonia solanaceum*, and *Geobacter sulfurreducens* was introduced in the analysis; however, it was trimmed from the tree to enhance visual resolution. Evolutionary analyses were conducted in MEGA7 [31].

The BvrR/BvrS operon was examined through bwa alignment [32] and SMALT v.0.5.8 mapping (http://www.sanger.ac.uk/resources/software/smalt/) in 126 *B. abortus* genomes to assess the presence and identity level of the genes included in the region, using as reference a region inferred from *B. suis* 1330. In addition, the number of SNPs, insertions, and deletions in each one of the genes was recorded manually (S3 Table).

## ChIP-Seq assay

The ChIP-Seq assay was performed as previously described [33] with the following modifications. The wild-type strain *B. abortus* 2308W and its derivative *bvrR*-mutant strain (negative control) were cultured until the mid-log phase in TSB. The bacterial cultures of each strain were divided into two equal parts. One part of each cultured strain was incubated for five minutes in a nutrient-limited medium (33 mM $KH_2PO_4$, 60.3 mM $K_2HPO_4$, 0.1% yeast extract) at pH 5.5 adjusted with citric acid. Those are stress conditions described previously for inducing BvrR phosphorylation [10]. The other part of the cultured strains was incubated in fresh TSB for 5 minutes as an additional control condition in which BvrR phosphorylation was expected to occur only transiently (rich conditions) [10]. Protein-DNA crosslinking was performed as indicated [33] and stopped by adding glycine to a final concentration of 125 mM, as described elsewhere [34]. After adding lysis buffer with lysozyme at 10 mg/ml, bacteria were lysed in the cell Disruptor Genie from Scientific Industries at 2800 rpm for 1 h, at 4°C, followed by overnight incubation with ChIP buffer at 37°C. The lysate was sonicated on ice (Branson Sonifier Digital cell disruptor S-450D 400W) by applying 25 bursts of 30 s at 30% amplitude and 30 s pause. A polyclonal rabbit anti-BvrR antibody [11] and magnetic beads coated with protein A were used for immunoprecipitation. DNA was extracted using a standard protocol of isopropanol precipitation [25]. Library construction and Illumina HiSeq 2500 HT sequencing were performed at Genomics Core Leuven, Belgium. The Bluepippin system (Sage Science) was used to select DNA fragments of approximately 220 bp that were sequenced paired-end.

Sequencing results were analyzed using Bioinformatics tools available on the Galaxy Project platform (https://usegalaxy.org/) [35]. The average and variance of reads per nucleotide were calculated in R Studio (http://www.rstudio.com) to establish a Z-score measured in terms of standard deviations from the mean, as described elsewhere [33, 36]. For each condition tested, ChIP-Seq signals considered significant were those that met all the following selection criteria: 1. To have a count of reads per nucleotide above the threshold (Z≥3), 2. To be absent in the negative control, and 3. To have a minimum length of seven consecutive nucleotides. Interactive visualization of the ChIP-Seq signals for the stress condition was constructed using the Bokeh Visualization Library (http://www.bokeh.pydata.org), and a custom Python code available at S1 File. Artemis [26] was used to look for the closest genes surrounding the significant signals. For significant ChIP-Seq signals located near the start of two divergent genes, both genes were considered *bonafide* putative BvrR target genes. For significant ChIP-Seq signals located within coding sequences, the corresponding gene and its adjacent downstream gene were considered *bonafide* putative BvrR target genes. The function annotation of all *bonafide*

putative BvrR target genes was manually curated using COG nomenclature and compared between conditions. The *bonafide* putative BvrR target genes found under stress conditions were used as input to perform an in-depth metabolic pathway analysis with BioCyc [37, 38] and manual curation. The DNA sequences of the significant ChIP-Seq signals were extracted from Artemis [26] and used as input for motif discovery with GLAM2 [39] to deduce a consensus sequence recognized by BvrR [40].

## Electrophoretic Mobility Shift Assay (EMSA)

Recombinant BvrR protein was produced and phosphorylated *in vitro* [11]. The direct interaction between BvrR-P and the upstream intergenic region of five selected target genes was analyzed by EMSA as described [10]. The selected target genes were: *tamA* (BAW_10045), *pckA* (BAW_12005), *bvrR* (BAW_12006), *omp25* (BAW_10696) and *virB1* (BAW_20068). The intergenic upstream regions of the 50S ribosomal protein L7/L12 gene *rplL* (BAW_11206) and *dhbR* (BAW_21104) were included as a negative control. DNA probes were prepared by PCR amplification of regions of ≈200 bp located upstream of the selected genes with primers listed in the S1 Table. The DIG Gel Shift 2nd Generation Kit" (Roche) was used for probe labeling, following manufacturer instructions. For EMSAs involving the *tamA*-probe, the probe was denatured 10 minutes at 95˚C before each assay. A 226 bp region from the *virB1* promoter known to bind directly to BvrR [11] was selected to analyze the described DNA binding motif. Ten overlapping oligonucleotides (S1 Table) covering this region were chemically synthesized (≈40 bp) (Invitrogen) and used as probes for EMSA as described [10]. Competitive EMSAs were performed as described [10]. Briefly, the digoxigenin-labeled probes tested in the direct EMSA for *tamA*, *omp25*, and *virB1* were incubated with BvrR-P (0.6 μM) and either an excess of the respective non-labeled probe as a competitor or separately, with an excess of non-labeled negative control probe (*rplL* or *dhb*) as a competitor. Samples were then processed as described for direct EMSAs.

## Mapping of transcriptional start sites

Total RNA was extracted from *B. abortus* 2308W as described above [11] and submitted to primer extension analysis according to a previously described protocol [41]. S1 Table lists the primers used for this purpose.

## Dnase I footprinting analysis of *virB* upstream region

The same 226 bp amplicon analyzed by EMSA was amplified using primer pvirdownI 5´-FAM labeled (S1 Table) and conditions described previously [42]. The fragment was gel purified with QiaQuick kit (Qiagen) and mixed with BvrR-P as described above for EMSA. The amplicon was digested using Dnase I and sequenced as described [43]. The bases protected from digestion were identified using Peak Scanner software from Applied Biosystems by superimposing the electropherograms of digested and non-digested DNA fragments.

## Results

### BvrR/BvrS and PTS^Ntr form an operon with cell cycle arrest, DNA repair, and stress response genes

In a previous study, co-transcription of *bvrR*, *bvrS*, and four downstream genes encoding a PTS^Ntr with regulatory functions was described in *B. melitensis* [21]. Thus, we first investigated if this transcriptional organization was conserved in *B. abortus* 2308W. Co-transcription of

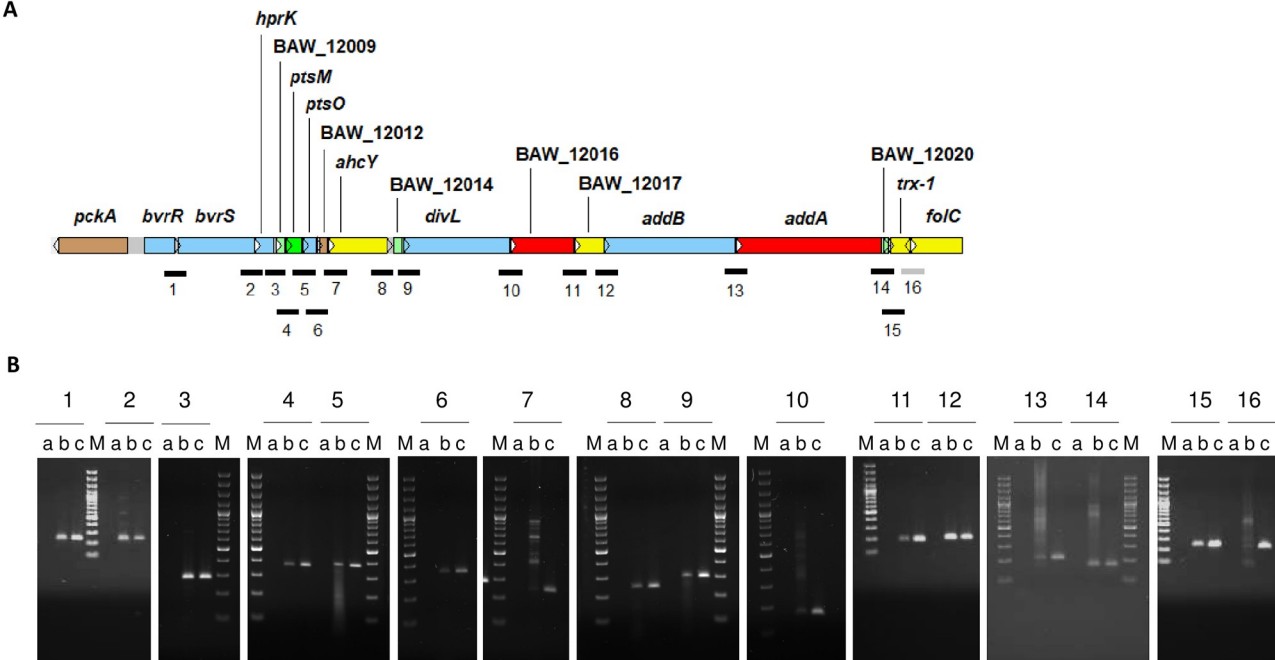

**Fig 1. Transcriptional organization of the *bvrR/bvrS* operon in *B. abortus* 2308W.** A. Schematic representation of the genomic region encoding the *bvrR/bvrS* operon (approximate coordinates in *B. abortus* 2308W genome: 2009267–2030918). The 5´-gene *pckA* was known to transcribe independently from *bvrR*, *bvrS*, and the PTS genes, unlike the 3´-genes BAW_12014 to *folC* [21]. The arrows indicate the orientation of transcription. The genes are color-coded according to their annotated general function: Brown = Pseudogenes and partial genes (remnants), Light blue = Regulators, Light green = Unknown, Dark green = Surface (inner membrane, outer membrane, secreted, surface structures), Yellow = Central/intermediary/miscellaneous metabolism, Red = Information transfer (transcription/translation + DNA/RNA modification). The lines below the genes illustrate the intergenic regions interrogated with primer pairs listed in S1 Table. The numbers1 to 16 follow their intergenic position along the operon. Black = co-transcribed regions as demonstrated by RT-PCR, Gray = non-co-transcribed regions as demonstrated by RT-PCR. B. Agarose gel electrophoresis of RT-PCR products obtained per region interrogated. Three lanes are shown for each RT-PCR result numbered from 1 to 16: a-minus RT (RNA, no RT), b-RT-PCR result and c-positive control (gDNA). The last five bands of the molecular marker (M) are 100, 200, 300, 400, and 500 bp-long. In total, 31 primer pairs were tested to span 16 overlapping regions of no more than 400bp. Only one representative RT-PCR product per region is shown. All amplicons were sequenced to corroborate their identity. The results shown correspond to the log phase of the growth curve in TSB and are also representative of the co-transcription events observed at the stationary growth phase in the same medium.

*bvrR*, *bvrS*, and 14 downstream genes was demonstrated by RT-PCR assays spanning intergenic regions and confirmed by Sanger sequencing of each obtained amplicon. (Fig 1).

Besides the PTS$^{Ntr}$, the downstream genes encode proteins/enzymes probably involved in functions related to shifting metabolic needs according to the environment, such as cell cycle arrest, LPS structure, DNA repair, gene recombination, and stress responses (Table 1). Synteny analysis of 126 whole-genome sequences of *B. abortus* strains showed the same organization and orientation as *B. abortus* 2308W operon and was consistent with *B. suis* 1330. The gene *pckA*, encoding a gluconeogenesis-essential PEP carboxykinase, was upstream and opposite the operon in the genomes analyzed. The gene is functional in *B. suis* 513 but not in *B. abortus* 2308W [44] and is consistently found as a pseudogene in the *B. abortus* genomes studied, with a premature stop codon in the same position. S3 Table describes additional detected SNPs in this region.

Conservation analysis of this region within representative *Alphaproteobacteria* genomes was performed. The overall structure of this operon is conserved in facultative intracellular/extracellular bacteria, practically absent in free-living bacteria, and absent in the strict intracellular animal pathogens (Fig 2). This configuration indicates that the operon was ancestral to *Alphaproteobacteria* members and subsequently reduced, translocated, or partially lost during

**Table 1. Description of the 16 genes that belong to *bvrR/bvrS* operon in *B. abortus* 2308W and correspondence with *B. abortus* 2308 genome.**

| Name(s) | Function according to genome annotation and literature |
|---|---|
| BAW_12006, *bvrR*, BAB1_2092 | Two-component transcriptional regulator BvrR [9] |
| BAW_12007, *bvrS*, BAB1_2093 | Two-component histidine kinase BvrS [9] |
| BAW_12008, *hprK*, BAB1_2094 | HPr kinase. Participates in the regulation of *B. melitensis* phospho-transfer system (PTS). The PTSNtr promotes the accumulation of a second messenger called (p)ppGpp in conditions of nitrogen starvation [21, 45] |
| BAW_12009, BAB1_2095 | Predicted protein with unknown function [24] |
| BAW_12010, *ptsM*, BAB1_2096 | PTS system fructose subfamily transporter subunit IIA [21, 24] |
| *ptsO* | NPr phosphocarrier protein:histidine phosphorylation site in HPr protein. Participates in *B. melitensis* PTS [21, 24] |
| BAW_12012, BAB1_2098 | Pseudogene. Frame shift and important deletion near 5' end; similar to BS1330_I2090 and BruAb1_2071 [24] |
| *ahcY* | Catalyzes the reversible hydrolysis of S-adenosylhomocysteine (SAH), producing homocysteine and adenosine. These compounds can be used as nitrogen sources during the intracellular life of *Brucella* spp [24, 46] |
| BAW_12014, BAB1_2100 | Predicted protein with unknown function [24] |
| BAW_12015, *divL* | Two-component sensor histidine kinase. Interacts with DivK and CCkA and controls the phosphorylation and proteolysis of CtrA [24, 47] |
| BAW_12016, BAB1_2102 | tRNA threonylcarbamoyladenosine biosynthesis protein TsaE. Participates in the processing of tRNA that read codons beginning with adenine [24] |
| BAW_12017, BAB1_2103 | Mannose-1-phosphate guanylyltransferase. Participates in amino sugar and sugar nucleotide metabolism (transferase activity). Could participate, redundantly with other genes, in the addition of mannose residues to LPS core structure, which helps to avoid the recognition by complement, antimicrobial peptides and pathogen recognition receptor complexes [24, 48, 49] |
| *addB* | ATP-dependent helicase/nuclease subunit B. Participates in DNA repair and recombination [24] |
| *addA* | Double-strand break repair helicase AddA. Also named ATP-dependent helicase UvrD/REP. AddA and AddB participate in the maintenance of DNA integrity during oxidative stress associated to a hostile intracellular environment [24, 36, 50] |
| BAW_12020, BAB1_2106 | Predicted protein with unknown function [24] |
| *trx-1* | Thioredoxin. Chaperones and folding catalysts. Participates in cell redox homeostasis and stress response. Trx-1 is differentially expressed in the attenuated strain *B. abortus* S19, which suggests it has a role in bacterial virulence [5, 24] |

the evolution of some groups. The absence of this operon in *Rickettsiae* and *Wolbachia* is commensurate with the genome reduction observed in these intracellular pathogens, experiencing drastic gene loss [51]. This absence reinforces the idea that the operon is required to transition from extracellular to intracellular environments.

## BvrR-P binds directly to genes related to virulence, cell envelope, energy metabolism, and cell division

TCS BvrR/BvrS is a gene master regulator essential for intracellular survival [9, 17]. It contributes to the metabolic fitness required to confront the different environments *Brucellae* encounter during host interaction [15, 16]. However, the genes directly regulated by TCS BvrR/BvrS, i.e., the direct regulon, are unknown. By ChIP-Seq analysis, we described and compared this regulon after exposure to conditions promoting BvrR phosphorylation, such as acidic nutrient-limited medium (stress conditions) and after exposure to rich medium at neutral pH (rich

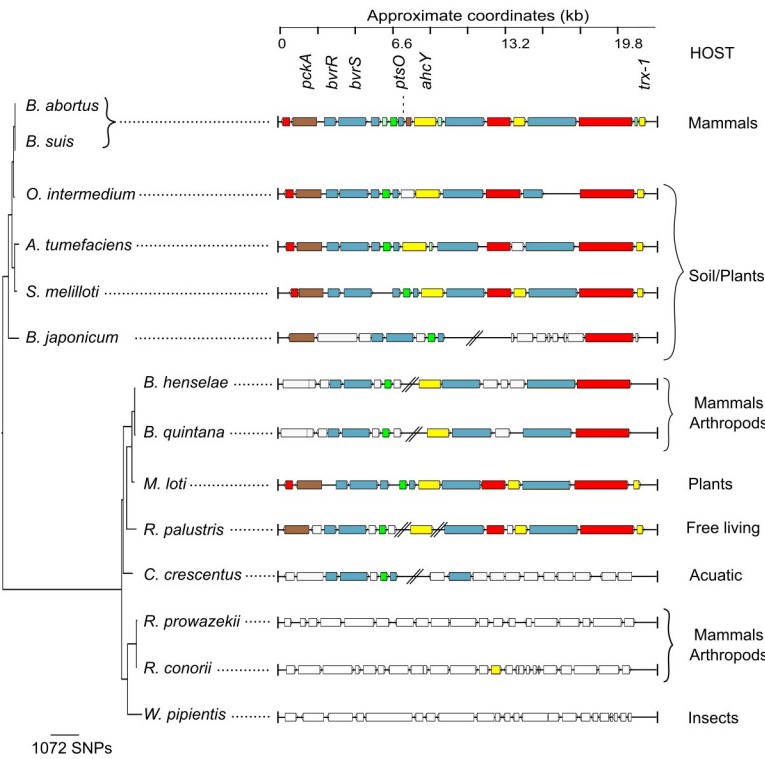

**Fig 2. BvrR/BvrS operon synteny analysis within representative *Alphaproteobacteria*.** Genes of *B. suis* 1330 were compared to representative genomes by BLAST and visualized through ACT. Blocks of different colors show the aligned coding sequences (CDs): grey—energy metabolism; red—information transfer (transcription/translation + DNA/RNA modification); green—surface (IM, OM, secreted, surface structures); pink -degradation of small molecules; light blue—regulators; orange—conserved hypothetical. CDs with no similarity with *B. suis* 1330 are white-colored. A Maximum Likelihood phylogenetic reconstruction based on the 16S rRNA gene was used to infer the evolutionary history of *Alphaproteobacteria*. The analysis involved 17 nucleotide sequences; the outgroup was trimmed from the tree to enhance resolution. There were a total of 1191 positions in the final dataset.

conditions). Sequencing results generated a sum of reads between 7.6 and 11.91 million per strain per condition tested. After trimming, 67.2–83.7% of the reads per sample were uniquely mapped, as expected for a ChIP-Seq experiment [52]. We corrected the noise signal background using a BvrR deficient strain under both tested conditions.

S1 File includes an interactive visualization of the obtained ChIP-Seq signals under stress conditions and according to the significance criteria described in Methods, after correcting for background noise. This stringent background noise correction lowered the absolute intensity signal; however, 321 ChIP-Seq signals were statistically significant: 63% in chromosome I and 37% in chromosome II (S4 Table). There was a five-fold increase in the number of significant ChIP-Seq signals under stress conditions compared to rich conditions. Analysis of the function category of the closest gene to a significant signal showed that the number of genes in all functional categories detected under rich conditions increased under stress conditions (Fig 3A and S4 Table), an observation that has also been described in other pathogens' TCSs [53]. Functions like energy metabolism and cell cycle control that were undetected under rich conditions appeared under stress conditions. This result suggests that the BvrR/BvrS direct regulon depends on external bacterial conditions. We consistently found significant ChIP-Seq signals surrounding 18 loci regardless of the condition tested (S4 Table). For example, we detected *btaE* encoding for an adhesion molecule required for full virulence and associated with a

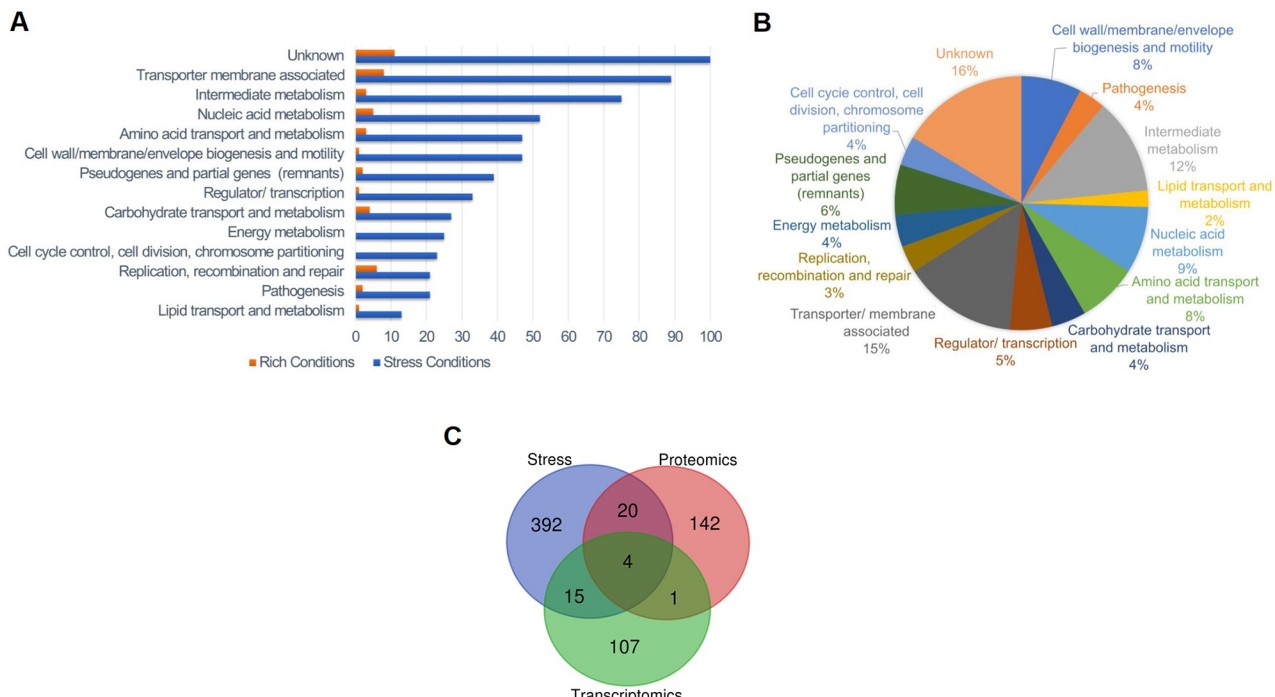

**Fig 3. Stress conditions trigger binding of BvrR discriminating according to function.** A. Distribution of the total number of significant signals according to the function of the closest gene and tested conditions shows a higher number of genes as BvrR putative targets under stress conditions and that targeting is selective according to function. Binding sites located close to rRNA genes were excluded from the analysis. Orange = Rich conditions. Blue = Stress conditions. B. Proportional distribution of BvrR binding sites according to the function of the closest gene in stress conditions. C. Venn Diagram generated with an online tool (https://bioinformatics.psb.ugent.be/webtools/Venn/) and illustrating the relationship between the BvrR target genes inferred in this study and the differentially expressed genes identified in previous proteomics [15] and transcriptomics [16] studies performed with the *B. abortus* 2308 WT and *bvrR*::*Tn5* strains.

specific adhesive pole in *B. suis* [54]. Other genes detected were the cyclic beta 1–2 glucan synthetase gene *cgs*, required for virulence in a mice model [55]; its upstream gene encoding an ABC transporter ATPase; and the *omp25* gene, already known to be regulated by BvrR/BvrS [12]. Additionally, the position of the significant signals related to each gene varied according to the external condition tested. In many cases, we found more than one signal close to the same gene (see below).

We detected a significant ($p \leq 0.05$) clustering of genes related to specific metabolic pathways in the dataset obtained under stress conditions (Fig 3B, Table 2). These include glutamate degradation, phospholipid biosynthesis (particularly the phosphatidylserine synthase pathway), UDP-glucose biosynthesis, LPS biosynthesis, and degradation of adenine and adenosine. The following carbohydrate related pathways had significant gene clustering: glycolysis variants, pentose-phosphate, erythritol, glucuronate interconversions, and gluconeogenesis from specific amino acids (e. g. arginine, cysteine, and glutamate). Several links between pathways related to energy metabolism and cell division are evident. GdhZ, part of the Gdh system (GdhZ/GdhA), is fundamental for the metabolic control of cell division [56]. The gene *dxs* encodes a 1-deoxy-D-xylulose-5-phosphate synthase linked to the pentose phosphate cycle and nucleotide synthesis and is upstream of *tlyA*, an rRNA methyltransferase related to *ftsJ* (BAW_20532). The regulation of *purA*, encoding an adenylosuccinate synthetase, by BvrR seems relevant since we detected five different signals surrounding this gene (Table 3). The enzyme uses aspartate as substrate, linking amino acid pathways to nucleic acid synthesis and cell division due to its clustering with *ade* (Table 2). The gene *omp16*, involved in the

**Table 2. Manually curated Biocyc enrichment analysis of metabolic pathways found in the BvrR ChIP-Seq under stress conditions.**

| Pathways | p-values | # of BvR binding sites | Genes found near the binding site | | | |
|---|---|---|---|---|---|---|
| | | | 2308W locus tag | 2308 locus tag | Name | Function |
| L-glutamate degradation | 0.009982516 | 1 | BAW_10217 | BAB1_0228 | gdhA | Glutamate dehydrogenase |
| | | 1 | BAW_11748 | BAB1_1827 | gdhZ | NAD-glutamate dehydrogenase |
| Phospholipid biosynthesis (Phosphatidylserine and phosphatidylethanolamine biosynthesis) | 0.009982516 | 1 | BAW_10452 | BAB1_0469 | psd | Phosphatidylserine decarboxylase |
| | | 1 | BAW_10453 | BAB1_0470 | pssA | CDP-alcohol phosphatidyltransferase |
| | | 1 | BAW_11116 | BAB1_1172 | ND | phosphatidate cytidylyltransferase |
| | | 3 | BAW_11911 | BAB1_1994 | ND | 1-acyl-sn-glycerol-3-phosphate acyltransferase |
| UDP-alpha-D-glucose biosynthesis | 0.009982516 | 1 | BAW_10055 | BAB1_0055 | pgm | Phosphoglucomutase catalyzes the interconversion of alpha-D-glucose 1-phosphate to alpha-D-glucose 6-phosphate |
| | | 1 | BAW_10301 | BAB1_0316 | pgi | Glucose-6-phosphate isomerase |
| | | 1 | BAW_11625 | BAB1_1702 | glmM | Phosphoglucosamine mutase, catalyzes the conversion of glucosamine-6-phosphate to glucosamine-1-phosphate. Peptidoglycan precursor biosynthesis. |
| | | 2 | BAW_20070 | BAB2_0070 | galU | UTP-glucose-1-phosphate uridylyltransferase |
| Glycolysis variants | 0.017889686 | 1 | BAW_10506 | BAB1_0525 | ppdK | Pyruvate phosphate dikinase |
| | | 1 | BAW_11576 | BAB1_1650 | rbsA-2 | Ribose import ATP-binding protein rbsA-2 xylitol transporter |
| | | 2 | BAW_11664 | BAB1_1741 | gap | Glyceraldehyde 3-phosphate dehydrogenase |
| | | 2 | BAW_11665 | BAB1_1742 | pgk | Phosphoglycerate kinase: G-protein beta WD-40 repeat |
| | | 1 | BAW_12010 | BAB1_2096 | ND | PTS system fructose subfamily transporter subunit IIA* |
| | | 2 | BAW_10366 | BAB1_0382 | ND | Cysteine desulfurase |
| | | 1 | BAW_20108 | BAB2_0109 | gnd | 6-phosphogluconate dehydrogenase, catalyzes the formation of D-ribulose 5-phosphate form 6-phospho-D-gluconate* |
| Thiazole biosynthesis I | 0.025337795 | 1 | BAW_10912 | BAB1_0951 | ND | Class V aminotransferase. Cysteine desulfurase. |
| | | 2 | BAW_10445 | BAB1_0462 | dxs | 1-deoxy-D-xylulose-5-phosphate synthase |
| Lipopolysaccharide biosynthesis | 0.025337795 | 1 | BAW_10036 | BAB1_0035 | kdsB | CMP-2-keto-3-deoxyoctulosonic acid synthetase, LPS biosynthesis (KDO)* |
| | | 1 | BAW_11115 | BAB1_1171 | lpxB | Lipid-A-disaccharide synthase |
| | | 1 | BAW_11116 | BAB1_1172 | ND | phosphatidate cytidylyltransferase |
| | | 2 | BAW_20204 | BAB2_0209 | waaA (kdtA) | 3-deoxy-D-manno-octulosonic acid transferase |
| Adenine and adenosine salvage | 0.027998101 | 5 | BAW_11618 | BAB1_1695 | purA | Adenylosuccinate synthetase, catalyzes the formation of N6-(1;2;-dicarboxyethyl)-AMP from L-aspartate; inosine monophosphateandGTP in AMP biosynthesis * |
| | | 1 | BAW_11903 | BAB1_1986 | hpt | Hypoxanthine phosphoribosyltransferase |
| | | 1 | BAW_20563 | BAB2_0587 | ade | Adenine deaminase regulator of chromosome condensation |

ND: not determined,

*manually inferred

invagination of the outer membrane during cell division [33], seems regulated by the TCS; this gene is next to *ftsH*, a protease upregulated during intracellular growth [57]. As anticipated, BvrR/BvrS seems to regulate other metabolic pathways related to membrane composition and virulence (Table 2) [16, 58, 59]. Fig 3C compares the results of this study and those reporting putative BvrR/BvrS targets, using transcriptomic and proteomic analysis of *B. abortus* 2308

and *bvrR* mutant strains. The three studies converged on identifying four common target genes, while our study compared only to proteomics or transcriptomics presented respectively 20 and 15 additional common target genes (S4 Table). Altogether, these results suggest that BvrR/BvrS TCS regulates crucial pathways vital for intracellular trafficking and survival. This is probably achieved by directly regulating enzymes located at crossroads or in tandem within these metabolic pathways [5, 57]. More work is needed to establish whether these *bonafide* BvrR-P binding sites are gene regulation sites.

We manually searched for genes relevant to intracellular survival [5, 16, 46, 75, 76] in this dataset and counted the significant ChIP-Seq signals near them. We identified 71 genes with one and up to five associated significant ChIP-Seq signals (S4 Table). We consider these genes as target genes putatively regulated by BvrR under stress conditions. Table 3 shows a selection of virulence-related genes. The genes with the highest number of binding sites detected (five) are involved in membrane transport and secretion (*virB1*, zinc, and arginine transport) and cell division. We found four binding sites in between *bvrR* and the upstream pseudogene *pckA*. Two genes, *bvrS* and *ftsY* (encoding part of the signal recognition particle), have three BvrR binding sites near them. Other genes related to the BvrR/BvrS and virulence such as *vjbR*, *virB2*, *virB4*, *virB5*, VirB effectors (*btpB*, *bspB*, *vceA*), *ptsH* genes related to flagella, and membrane-associated genes such as *omp25*, *omp16*, *tamA*, *tamB*, *sagB*, and *btaE*, have two or one binding sites near them. Genes related to cell cycle, receptor binding, erythritol metabolism, nickel, manganese, and magnesium transport showed one or two BvrR-P binding sites. Additional genes related to virulence, such as *cgs*, *ppdK*, and *ureC* had one binding site [44, 55, 57, 68, 77]. These observations suggest that the regulation of virulence genes is complex and that bacterial transcription factors do not always behave as per the textbook operon model. The interactions between BvrR-P and its binding sites probably depend on the concentration of BvrR-P at a given moment. The involvement of additional transcription factors might also be possible, as described for *virB* (see below).

To confirm that genes *tamA*, *pckA*, *bvrR*, *omp25*, and *virB1* are direct BvrR-target genes, we evaluated by EMSA the interaction of DNA fragments encoding their upstream intergenic regions with recombinant *in vitro* BvrR-P. As shown in Fig 4A, incubation of BvrR-P with these DNA probes retarded the electrophoretic migration pattern compared to identical probes incubated without the BvrR-P, indicating DNA-protein interaction. We did not observe differences in the migration pattern of the two DNA probes selected as negative controls (*rplL* and *dhbR*). The interaction was specific because an excess of unlabeled control probes did not alter the migration pattern, compared to experiments with an excess of unlabeled target probes (Fig 4B).

In S4 Table, we have included the information obtained after mapping the transcriptional start sites (TSS) of target genes confirmed by EMSA and previously known information about additional TSS. For example, the TSS of *tamA* is downstream of the EMSA binding site, and in the case of *omp25* and *virB1*, the TSS is within the EMSA binding site. Furthermore, through ChIP-Seq, we detected five binding sites related to *virB* (Table 3 and Fig 5F), suggesting that additional TSS located within the coding region could function under different conditions [8, 79].

## BvrR recognizes a consensus sequence of at least 14 bp nucleotides long

The sequences of the significant ChIP-Seq signals were used as input for motif discovery with GLAM2 [39] to unveil a possible DNA primary structure pattern recognized by BvrR-P under conditions that mimic the intracellular environment. The 14 nucleotides long DNA binding motif is depicted in Fig 5A, with the last six nucleotides matching a previously reported *in silico*

**Table 3. Selection of manually curated genes of interest according to number of significant signals close or within their CDS.**

| 2308W | 2308 | Name | Function |
|---|---|---|---|
| **Genes with 5 binding sites** | | | |
| BAW_11618 | BAB1_1695 | *purA* | Adenylosuccinate synthetase, important for virulence [17] |
| BAW_20068 | BAB2_0068 | *virB1* | Type IV secretion system protein VirB1, important for virulence, important for virulence [60] |
| BAW_11934 | BAB1_2018 | *zntR* | Zn responsive regulator of zntA, important for virulence [61] |
| BAW_11935 | BAB1_2019 | *zntA* | Zn exporter, important for virulence [61] |
| BAW_11873 | BAB1_1956 | BAW_11873 | ABC transporter permease binding-protein dependent transport system inner membrane protein |
| BAW_11874 | BAB1_1957 | BAW_11874 | Arginine ABC transporter ATP-binding protein |
| **Genes with 4 binding sites** | | | |
| BAW_12005 | BAB1_2090 -BAB1_2091 | *pckA* | Pseudogene. Premature stop codon. Similar to BS1330_I2083 and BruAb1_2064; phosphoenolpyruvate carboxykinase (ATP). A *pckA* mutation in *B. abortus* 2308 has no effect in Raw 264.7 macrophage intracellular replication and is not attenuated in the mice model |
| BAW_12006 | BAB1_2092 | *bvrR* | Two-component transcriptional regulator BvrR, important for virulence [9] |
| **Genes with 3 binding sites** | | | |
| BAW_11853 | BAB1_1934 | *ftsY* | Cell division protein |
| BAW_12007 | BAB1_2093 | *bvrS* | Two-component histidine kinase BvrS, important for virulence [9] |
| **Genes with 2 binding sites** | | | |
| BAW_20152 | BAB2_0156 | *flgH* | Flagellar basal body L-ring protein, important for virulence [62] |
| BAW_21057 | BAB2_1103 | *motB* | Flagellar motor protein MotB, important for virulence [62] |
| BAW_10069 | BAB1_0069 | *btaE* | Hyaluronate-binding autotransporter adhesin required for virulence, important for virulence [54] |
| BAW_10727 | BAB1_0756 | *btpB* | VirB type IV secreted effector, important for virulence [63] |
| BAW_20067 | BAB2_0067 | *virB2* | Type IV secretion system protein VirB2, important for virulence [60] |
| BAW_20116 | BAB2_0118 | *vjbR* | LuxR family regulatory protein VjbR, important for virulence [64] |
| BAW_20365 | BAB2_0377 | *eryG; rbsB-2* | Erythritol periplasmic binding protein, important for virulence [65] |
| BAW_20366 | BAB2_0378 | *deoR* | DeoR family regulatory protein, erythritol regulator [66] |
| BAW_20415 | BAB2_0432 | *nikR* | Nickel-responsive regulator of nikA, nikB, nikC, nickD and nikE |
| BAW_20417 | BAB2_0435 | *nickB* | Nickel transporter permease NikB |
| **Genes with 1 binding site** | | | |
| BAW_10106 | BAB1_0108 | *cgs* | Cyclic beta 1–2 glucan synthetase, important for virulence [67] |
| BAW_10506 | BAB1_0525 | *ppdK* | Pyruvate phosphate dikinase, important for virulence [68] |
| BAW_20108 | BAB2_0109 | *gnd* | 6-phosphogluconate dehydrogenase, catalyzes the formation of D-ribulose 5-phosphate form 6-phospho-D-gluconate |
| BAW_11379 | BAB1_1445 | *ftsA* | Cell division protein FtsA. Is involved in the assembly of the Z ring. May serve as a membrane anchor for the Z ring |
| BAW_11380 | BAB1_1446 | *ftsQ* | Cell division protein FtsQ |
| BAW_10696 | BAB1_0045 | *tamA* | Bacterial surface antigen (D15), important for virulence [17] |
| BAW_10045 | BAB1_0722 | *omp25* | Outer membrane protein Omp25 precursor |
| BAW_11648 | BAB1_1725 | *motA* | Flagellar motor protein MotA, important for virulence [69] |
| BAW_11649 | BAB1_1726 | ND | COG1360 Flagellar Motor Protein |
| BAW_21059 | BAB2_1105 | *fliF* | Flagellar MS-ring protein, the MS-ring anchors the flagellum to the cytoplasmic membrane; part of the flagellar basal body which consists of four rings L; P; S; and M mounted on a central rod, important for virulence [62] |
| BAW_11312 | BAB1_1378 | *ureC* | Urease subunit alpha |
| BAW_11626 | BAB1_1703 | *ftsH* | ATP-dependent zinc metalloprotease |
| BAW_11630 | BAB1_1707 | *omp16* | Pal = peptidoglycan-associated lipoprotein; also called omp16 |
| BAW_10046 | BAB1_0046 | *tamB* | Autotransporter translocation and assembly factor TamB This protein translocates adhesins and is essential for full virulence and intracelullar trafficking, important for virulence [70] |
| BAW_10686 | BAB1_0712 | *bspB* | Type IV secretion effector. This protein mediates inhibition of host secretion, important for virulence [71] |
| BAW_11394 | BAB1_1460 | *mntH* | Manganese transport protein MntH, important for virulence [72] |
| BAW_11577 | BAB1_1652 | *vceA* | VirB type IV secreted effector vceA, important for virulence [73] |
| BAW_20064 | BAB2_0064 | *virB5* | Type IV secretion system protein VirB5, important for virulence [60] |

*(Continued)*

**Table 3.** (Continued)

| 2308W | 2308 | Name | Function |
|-------|------|------|----------|
| BAW_20065 | BAB2_0065 | *virB4* | Type IV secretion system protein VirB4, important for virulence [60] |
| BAW_12010 | BAB1_2096 | BAW_12010 | PTS system fructose subfamily transporter subunit IIA |
| BAW_12011 | BAB1_2097 | *ptsH* | Phosphocarrier HPr protein:histidine phosphorylation site in HPr protein |
| BAW_20348 | BAB2_0360 | *mgtE* | Divalent cation transporter/ magnesium transporter MtgE |
| BAW_20420 | BAB2_0438 | *nikE* | Nickel transporter ATP-binding protein NikE with NikABCD is involved with nickel transport into the cell |
| BAW_21102 | BAB2_1150 | *bhuA* | TonB-dependent receptor protein, important for chronic infection in mice [74] |

Purple: Cation metabolism related

Green: BvrR/BvrS-VjbR-VirB related

Blue: Cell cycle control, cell division, chromosome partitioning

Gold: Receptor binding related

Grey: Erythritol metabolism

sequence prediction recognized by BvrR [81]. We decided to further analyze and contextualize this finding in the most characterized *Brucella* virulence factor, the T4SS *virB*. According to our data, BvrR-P binds by EMSA to the previously characterized *virB1* promoter region [42] located in a 226 bp fragment upstream of the start codon. None of the significant ChIP-Seq signals were detected in this region. However, four nucleotide sequences within this 226 bp fragment are similar to the DNA binding motif (orange triangles in Fig 5F). Hence, we decided to

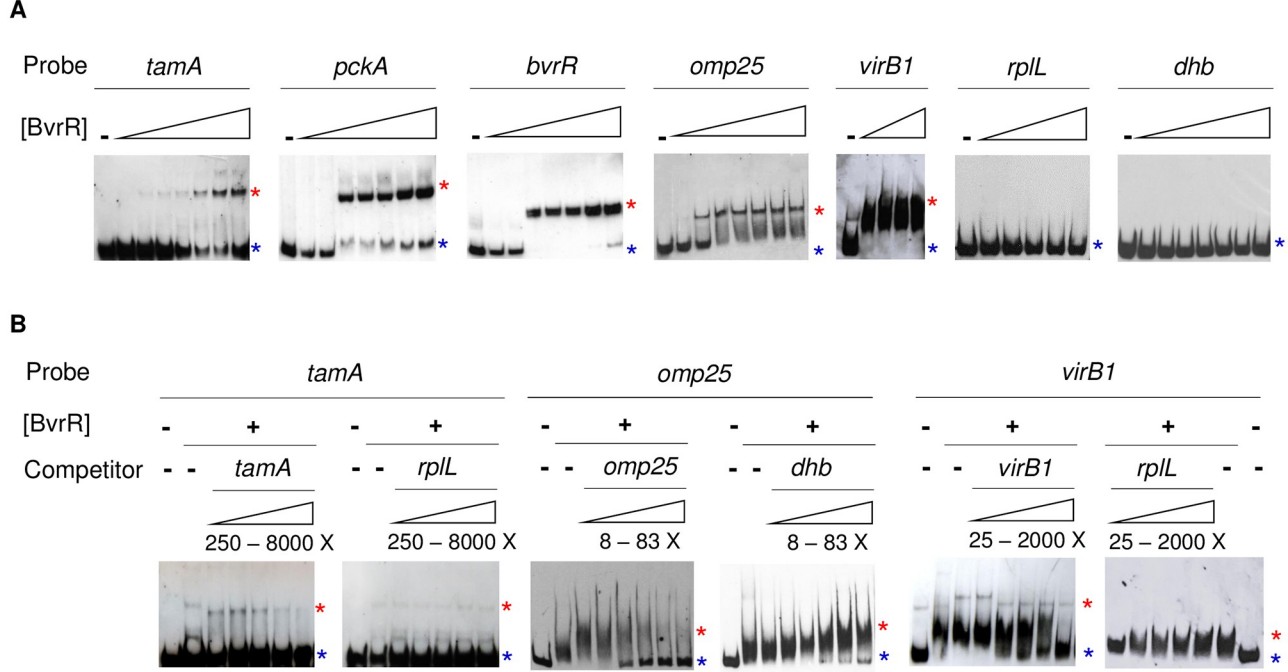

**Fig 4. Biochemical confirmation of the direct binding of BvrR-P to the upstream intergenic region of selected target genes.** A. Direct EMSA with the following probes: *tamA*, *pckA*, *bvrR*, *omp25*, *virB1*, and negative controls *rplL* and *dhbR*. The probes were designed based on the location of the significant ChIP-Seq signals obtained in this study and previous information about transcriptional units and promoter structures when available [24, 42, 78]. Protein concentrations for each experiment varied from 0.05 to 2.6 μM. B. Competitive EMSA using increasing concentrations of free probe (competitor). The protein concentration was 0.6 μM. Red asterisks represent the migration pattern of a protein-DNA complex (shift). Blue asterisks represent the migration pattern of a free probe (no shift). Experiments in panels A and B are independent of each other. All gels have either negative (probe without protein) or positive controls (probe with protein) to compare. These results are representative of at least three independent experiments.

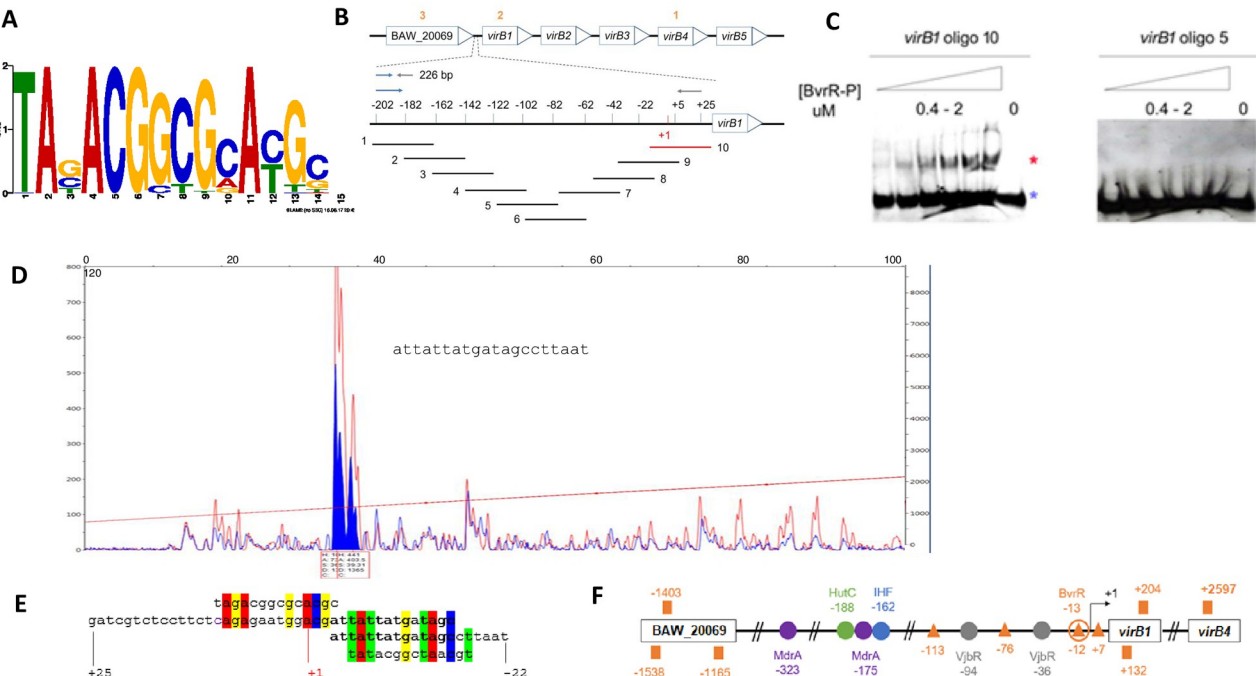

**Fig 5. Location of the DNA binding motif recognized by BvrR-P in the promoter region of *virB1*.** A. DNA binding motif recognized by BvrR-P under conditions that mimic the intracellular environment obtained using GLAM2 for motif discovery within the significant ChIP-Seq signals dataset. B. Schematic non-scale representation of the *virB* operon and upstream gene BAW_20069. Orange numbers represent each of the three significant ChIP-Seq signals in this region. The zoomed area represents a 226bp *virB1* upstream region, with a promoter region previously reported [42]. Red and black lines represent oligonucleotides that did and did not interact with BvrR-P by EMSA. C. EMSA results for the *virB1*-oligonucleotides number 10 and 5. Red and blue asterisks represent the migration patterns of a protein-DNA complex (shift) and a free probe (no shift). D. Dnase I footprinting results using BvrR-P and the 226 bp DNA fragment encoding the *virB1* promoter region. The traces are the Dnase I digested DNA fragments incubated with bovine serum albumin as control (red) or with BvrR (blue). Blue-filled peaks represent the Dnase I-protected region inferred according to [43]. The panel includes the DNA sequence obtained after Sanger sequencing. E. Sequence alignment between two configurations of the DNA binding motif (first and last line) derived from the motif shown in Fig 5A, the *virB1*-oligonucleotide 10 (second line), and the Dnase I-protected region (third line). F. BvrR-P binding sites in the *virB* regulatory region. The orange squares represent the location of ChIP-Seq signals obtained in this study. The orange triangles represent the location of regions with percentages of sequence similarity to the DNA binding motif shown in A, ranging from 50 to 71.43%. The orange circle represents the BvrR-P binding site confirmed by EMSA and DnaseI footprinting. The colored circles (purple, green, blue, and gray) represent the binding sites for other transcription factors described to regulate the expression of the *virB* operon [80]. Number coordinates are relative to the transcription start site (black arrow). These results are representative of at least three independent experiments.

design and test by EMSA ten overlapping oligonucleotides (Fig 5B) encompassing the 226 bp intergenic upstream region of *virB1*, previously used as a probe (Fig 4A). *VirB1*-oligonucleotide number 10 was the only one interacting with BvrR-P (Fig 5C). Dnase I footprinting analysis using BvrR-P and the *virB1* upstream 226 bp fragment confirmed BvrR-P binding (Fig 5D). The protected sequence partially matches *virB1*-oligonucleotide number 10 (Fig 5E, third and second lanes). Additionally, it contains one of the nucleotide sequences with similarity (71.43%) to the DNA binding motif (Fig 5E, third and fourth lanes). Next to the protected sequence, another DNA fragment showed 50% sequence similarity to the DNA binding motif (Fig 5E, second and first lanes), suggesting that alternative DNA binding sites with different affinities might be available under different environmental conditions. Fig 5F shows a schematic representation of the location of the five ChIP-Seq signals found within the vicinity of *virB1* and BAW_20069, the four putative DNA binding motifs found within the *virB* promoter, and the BvrR-P binding site confirmed by EMSA and Dnase I footprinting. Fig 5F also shows the location of other binding sites for different transcription factors previously shown to regulate the expression of the *virB* operon. As shown, the ChIP-Seq signals did not match

the location of the biochemically confirmed BvrR-P binding site. The reason why the described binding site found by EMSA and Dnase I footprinting was not detected by ChIP-Seq remains elusive. However, the experimental conditions for EMSA and Dnase I footprinting are by principle different than those of ChIP-Seq. These results contribute to highlighting the complexity of the *virB* promoter fine-tuning expression.

## Discussion

In *B. abortus*, the ability to sense environmental changes through BvrR/BvrS TCS is transcriptionally linked to strategic functions for successful trafficking and survival in different milieus [5, 10, 16, 45, 48–50]. Dozot and collaborators demonstrated that in *B. melitensis*, *bvrR* and *bvrS* were transcriptionally coupled to a downstream encoded PTS$^{Ntr}$ with regulatory functions [21]. Our results indicate that BvrR/BvrS TCS is co-transcribed with 14 additional genes in *B. abortus*, independently of the growth stage. The synteny and organization of this operon in some *Alphaproteobacteria* members closely related to *Brucella* suggest that this region is responsible for coordinating the expression of carbon and nitrogen metabolic pathways, according to the energy sources and environmental conditions found during events leading to host association. This observation agrees with recent reports [20].

Orthologs to BvrR/BvrS TCS described in the *Alphaproteobacteria*, required for pathogenic/symbiotic lifestyles, sense and respond to local conditions associated with their specific environments. For example, BatR/BatS from *Bartonella* spp. senses the physiological pH of the mammalian blood (pH 7.4), discriminating the host environment from the arthropod vector and regulating the expression of several virulence genes, like the T4SS VirB and its effectors [82]. In *A. tumefaciens*, ChvG/ChvI is essential for membrane integrity, virulence, and bacterial growth under acidic conditions [83]. In the plant endosymbiont *S. meliloti*, the TCS ExoS/ChvI is essential to establish endosymbiosis [84, 85]. Interestingly, this transcriptional organization is not conserved in all the *Alphaproteobacteria*, which correlates with the group evolution and lifestyles. Intracellular bacteria associated with invertebrates, animals, humans, or both have evolved by gene loss, such as *Bartonella* spp., *Rickettsia* spp., and *Wolbachia* spp., intracellular pathogens with smaller genomes than *Brucella* spp., a facultative intracellular extracellular parasite [10, 86].

During intracellular stages, it is likely that *B. abortus* grows using a low oxygen tension type of respiration with a rate reduction of central carbon metabolic pathways, such as the TCA cycle, the pentose phosphate pathway, and decrease of periplasmic transporters. When sugar supplies are limited, the bacterium switches to anaplerotic routes increasing amino acid catabolism. Therefore, glutamate fuels the TCA as an energy source, and role of the glyoxylate shunt is minimum [5, 46, 76]. As in *Rhizobium* species, glutamate could be used as carbon, nitrogen, and energy source. This double role of glutamate can be explained by its connection with the TCA cycle, gluconeogenesis, and the urea cycle [46, 56].

The TCS BvrR/BvrS could regulate these energy pathways in a temporal and spatial simultaneous fashion since most of the described target genes are at these pathways' crossroads or in tandem. The proposed model of energy influx [76, 87] agrees with our results (Fig 6). Due to a lack of phosphofructokinase, glycolysis is unlikely to be active in *B. abortus*. Accordingly, in zoonotic *Brucella*, the pentose phosphate pathway fuels the TCA cycle for glucose oxidation [88, 89]. Phosphorylated glucose enters the pentose cycle to produce glyceraldehyde-3-phosphate, channeled into the TCA via pyruvate. The TCS BvrR/BvrS also controls sugar transporters (PTS) genes as well as the metabolism of erythritol and xylitol. These monosaccharides can enter the pentose phosphate pathway. The energy pathways under BvrR/BvrS control probably include anaplerotic routes using glutamate and other glucogenic amino acids (arginine,

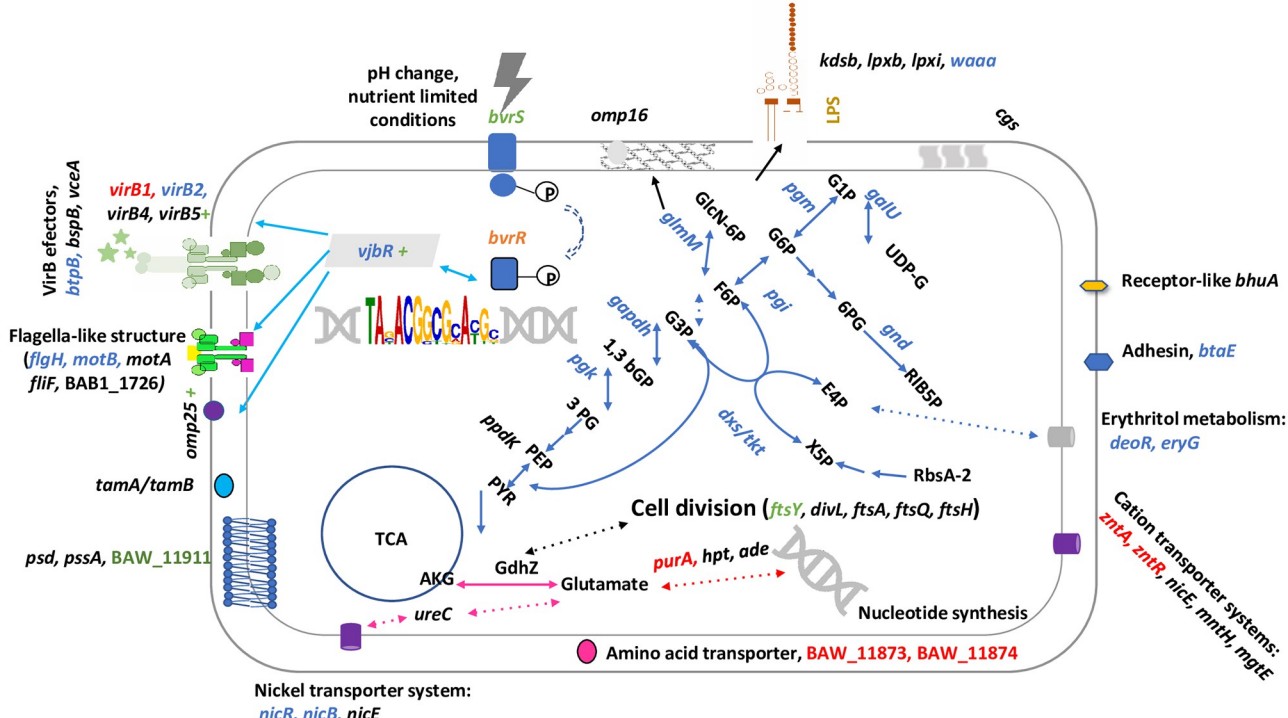

**Fig 6. Putative BvrR/BvrS TCS regulon in *B. abortus* upon entry into the host cell.** The recognition of the intracellular environment-associated cues by BvrS induces its autophosphorylation and consequently BvrR phosphorylation. As a result, BvrR-P increases its affinity for nucleotide regions similar to the consensus sequence, including its operon and regions encoding crossroad enzymes crucial for metabolic pathways required for intracellular trafficking. Genes according to the number of binding sites to BvrR-P found by ChIP-Seq are shown. Gene names with one binding site are in black, two binding sites: blue, three binding sites: green, four binding sites: orange, and five binding sites: red. Blue arrows indicate part of the energy metabolism, mainly the pentose phosphate cycle, erythritol metabolism, and Entner–Doudoroff pathway. The connection between the TCA cycle and nitrogen metabolism is represented with pink arrows. Light blue arrows represent VjbR targets. The green plus sign indicates positive regulation exerted by BvrR/BvrS. Pointed arrows indicate pathways partially represented.

cysteine). The evidence presented here also indicates that TCS BvrR/BvrS synchronizes this energy flow with vital functions, such as cell division, nucleic acid synthesis, and cell envelope homeostasis (Fig 6).

During intracellular life, *Brucella* requires prototrophy and tight regulation of nitrogen sources available for metabolism [46]. In agreement with those requirements, BvrR/BvrS could regulate the adenine and adenosine salvage pathways. The fact that five significant signals were found related to *purA*, encoding an adenylosuccinate synthetase, relating nucleic acid and amino acid biosynthesis pathways, suggests that the regulation is relevant and probably exerted at different growth conditions, as previously suggested for this enzyme [17]. BvrR/BvrS probably regulates enzymes involved in the biosynthetic pathways of phosphatidylserine and phosphatidylethanolamine. This finding is consistent with the peculiar cell envelope phospholipid composition required for full virulence [46, 58]. However, no differences in phospholipid profiles were observed between *B. abortus* wild type and *bvrR* or *bvrS* mutant [14]. We also found several genes encoding functions related to LPS biosynthesis as direct BvrR-P targets. Studies have demonstrated that the stealthy pathogenic strategy of *Brucella* is related to the structure of its LPS [49, 90]. Interestingly, *galU* encoding a UTP-glucose-1-phosphate uridylyltransferase and *pgm* encoding phosphoglucomutase are in the same metabolic crossroad (Fig 6). They participate in the biosynthesis of glucose-1-phosphate, the activated form of glucose required to synthesize polysaccharides, glycoproteins, and glycolipids. A *pgm* mutant carries

an incomplete LPS core and is defective in synthesizing periplasmic β-glucans [59, 91]. Expression of this enzyme increases when erythritol is present and is highly increased during intracellular *Brucella* stages [5].

Genes related to virulence show direct binding to BvrR-P, including the BvrR/BvrS operon itself. Some new BvrR-P virulence-related targets are unveiled, reinforcing the idea that BvrR/BvrS coordinates the expression of virulence traits according to environmental signals. Some of these targets were selected to confirm the direct and specific binding of BvrR-P to their upstream regions by EMSA and Dnase I footprinting. Transcription of the T4SS VirB is tightly controlled as *Brucella* transit in different intracellular compartments [4, 92]. Several regulators have been implicated in the expression of the *virB* operon besides BvrR [34, 42, 93–96], suggesting that the expression of *virB* is regulated through an intricate regulatory network as depicted in Fig 5F. Five putative binding sites were detected related to *virB* by ChIP-Seq. Additionally, four regions with sequence similarity to the DNA binding motif were identified within the *virB1* promoter, and one of them was confirmed as a BvrR-binding site by biochemical means. These binding sites show different degrees of similarity to the obtained consensus sequence, suggesting that the affinity of BvrR-P for these sites might play a role in controlling *virB* transcription in coordination with other regulatory molecules previously described [75]. The detection of more than one significant ChIP-Seq signal close to a gene, including some within the coding region and even at the 3´ end of the coding region, is opening possibilities for further research related to the role of antisense transcription [97] within the BvrR/BvrS regulon, the presence of gene overlapping [98] and non-conventional promoter structure [99] in *Brucella*. To our knowledge, transcription could be promoted from unusual sites, and multiple binding sites could be needed for optimal binding. Some activators are known to bind to unusual regions and induce promoter activity, as it has been described for other bacterial pathogens [100, 101]. PhoP of *B. subtilis* is a response regulator for phosphate starvation, which induces activation of *pstS* by binding to an upstream region (-40 to -132) and a coding region (+17 to +270) required for complete promoter activity. In addition, the coding region box had a low affinity for PhoP-P, suggesting a dynamic DNA-protein binding, in which the regulator is required to start transcription [102]. Global regulators are known to bind to a collection of sites, and the regulatory effect on each binding site would be dependent on the protein concentration at any given moment, its affinity, and additional transcription factors. Hence, they can be activators, repressors, have dual regulatory roles, or have no described regulatory function [103–107].

In *Salmonella*, the global response regulator OmpR activates the expression of SsrAB, a two-component system located on the pathogenicity island 2 (SPI-2). Several OmpR binding sites were found upstream of *ssrA* and upstream and within the *ssrB* coding sequence [108–110].

The BvrR binding sites described in this work should be considered *bonafide* putative gene regulation sites. Some of these regions have been previously identified as putative BvrR/BvrS targets [15, 16] and deserve further investigation. Additionally, to our knowledge, very few *Brucella* promoter regions have been functionally characterized; hence, this essential information to unveil the mechanisms of gene regulation is missing. In this sense, confirmation of the role of each BvrR-P binding site, by itself or in combination with other BvrR-P binding sites or additional regulatory mechanisms, as well as gene promoter characterization, certainly will shed some light on understanding this complex phenomenon.

The work presented here helps to understand how a conserved TCS contributes to the dynamics and complex gene regulatory functions during host-microbe interactions. BvrR/BvrS seems to contribute to metabolic fitness at several levels: i) regulating specific carbon and nitrogen pathways via interaction with a Pts$^{Ntr}$ co-transcribed system, ii) by direct interaction

with genetic regions coding for enzymes located at the crossroads of these specific pathways, iii) possible interaction of different BvrR-P binding sites according to BvrR-P concentration and presence of additional transcriptional factors known to be involved in this process and iv) adjusting the target genes according to the external bacterial conditions. Further work is needed to understand the role of BvrR/BvrS in the *Brucella* life cycle. Similar metabolic controls might be present in other *Alphaproteobacteria* living in close association with cells.

## Supporting information

**S1 Raw images. Raw gels and EMSAs from this study.**
(PDF)

**S1 Table. List of oligonucleotides used in this study.**
(PDF)

**S2 Table. Genomes used for α-*Proteobacteria* phylogenetic reconstruction.**
(PDF)

**S3 Table. WGS-Metadata_operon.**
(XLSX)

**S4 Table. ChIP-seq analysis results.**
(XLSX)

**S1 File. Interactive visualization of all the ChIP-seq signals obtained under stress conditions.** The X-axis displays the number of reads per nucleotide, and the Y-axis shows the coordinates of the *Brucella abortus* 2308 W genome (3.26 Mb). Significant signals are colored in yellow and non-significant signals in blue. Significant signals have reads per nucleotide above the threshold ($Z \geq 3$), are absent in the BvrR-deficient strain (negative control), and have a minimum length of seven consecutive nucleotides. The graphic below displays all the signals in a condensed fixed version, while the graphic above zooms in the region delimited by the selection box. The selection box is represented in the graphic below by a blue rectangle surrounded by a dotted line. To zoom other regions, drag the selection box's middle or edges in the graphic below. As shown in this visualization, having a negative control allowed us to confidently discriminate between significant and non-significant signals, regardless of their Z-score or size in nucleotides. We recommend visualization using Mozilla Firefox.
(ZIP)

## Acknowledgments

We thank Reynaldo Pereira-Reyes for his technical assistance with Dnase I footprinting and the purification of recombinant BvrR for EMSA experiments. We also thank Gustavo Segura-Umaña for his help with the interactive visualization of the genomic regions considered to be bound to BvrR under stress conditions.

## Author Contributions

**Conceptualization:** Xavier De Bolle, Caterina Guzmán-Verri.

**Data curation:** Olga Rivas-Solano, Mathilde Van der Henst, Amanda Castillo-Zeledón, Marcela Suárez-Esquivel, Lohendy Muñoz-Vargas, Zeuz Capitan-Barrios, Caterina Guzmán-Verri.

**Formal analysis:** Olga Rivas-Solano, Mathilde Van der Henst, Amanda Castillo-Zeledón, Marcela Suárez-Esquivel, Lohendy Muñoz-Vargas, Zeuz Capitan-Barrios, Caterina Guzmán-Verri.

**Funding acquisition:** Olga Rivas-Solano, Nicholas R. Thomson, Esteban Chaves-Olarte, Edgardo Moreno, Xavier De Bolle, Caterina Guzmán-Verri.

**Investigation:** Olga Rivas-Solano, Mathilde Van der Henst, Amanda Castillo-Zeledón, Marcela Suárez-Esquivel, Lohendy Muñoz-Vargas, Zeuz Capitan-Barrios, Caterina Guzmán-Verri.

**Project administration:** Xavier De Bolle, Caterina Guzmán-Verri.

**Resources:** Nicholas R. Thomson, Esteban Chaves-Olarte, Edgardo Moreno, Xavier De Bolle, Caterina Guzmán-Verri.

**Supervision:** Xavier De Bolle, Caterina Guzmán-Verri.

**Validation:** Olga Rivas-Solano, Mathilde Van der Henst, Amanda Castillo-Zeledón, Marcela Suárez-Esquivel, Lohendy Muñoz-Vargas, Zeuz Capitan-Barrios, Caterina Guzmán-Verri.

**Visualization:** Olga Rivas-Solano, Amanda Castillo-Zeledón, Marcela Suárez-Esquivel.

**Writing – original draft:** Olga Rivas-Solano, Amanda Castillo-Zeledón, Caterina Guzmán-Verri.

**Writing – review & editing:** Olga Rivas-Solano, Mathilde Van der Henst, Marcela Suárez-Esquivel, Lohendy Muñoz-Vargas, Zeuz Capitan-Barrios, Nicholas R. Thomson, Esteban Chaves-Olarte, Edgardo Moreno, Xavier De Bolle, Caterina Guzmán-Verri.

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
