## [Decision Letter · Decision Letter 0]

27 Apr 2022

PONE-D-22-09403The regulon of Brucella abortus two-component system BvrR/BvrS reveals the coordination of metabolic pathways required for intracellular lifePLOS ONE

Dear Dr. Guzman-Verri,

Thank you for submitting your manuscript to PLOS ONE. After careful consideration, we feel that it has merit but does not fully meet PLOS ONE’s publication criteria as it currently stands. Therefore, we invite you to submit a revised version of the manuscript that addresses the points raised during the review process.

Sorry for the delay in getting your paper reviewed!  As you can see from their comments, both reviewers feel that the manuscript presents information that will be important to the field. But both also have concerns about the presentation that will need to be corrected/clarified before the paper will be considered suitable for publication in *PLOS ONE*. A particular concern that was raised was that in some cases genes are being identified as targets of  BvrR regulation by ChIPseq analysis without independent verification that BvrR actually regulates these genes.

Considering the reviewers' evaluations, I am going to ask that you submit a revised version of  the manuscript that adequately and appropriately addresses all of  the concerns raised by both reviewers. 

We look forward to receiving your revised manuscript.

Kind regards,

Marty Roop

Academic Editor

*PLOS ONE*

Journal Requirements:

Reviewers' comments:

Reviewer's Responses to Questions

**Comments to the Author**

1. Is the manuscript technically sound, and do the data support the conclusions?

Reviewer #1: Partly

Reviewer #2: Partly

2. Has the statistical analysis been performed appropriately and rigorously? 

Reviewer #1: Yes

Reviewer #2: I Don't Know

3. Have the authors made all data underlying the findings in their manuscript fully available?

Reviewer #1: Yes

Reviewer #2: Yes

4. Is the manuscript presented in an intelligible fashion and written in standard English?

Reviewer #1: Yes

Reviewer #2: Yes

5. Review Comments to the Author

Reviewer #1: The manuscript by Rivas-Solano and colleagues describes the regulatory link between the BvrR/BvrS two-component system and metabolism in Brucella abortus. The authors have a long history with this system, including the identification of BvrR/BvrS and the genetic and biochemical characterization of the system, and this manuscript expands on that work to demonstrate the role that BvrR/BvrS plays in controlling the expression of genes related to metabolism. The bvrR/bvrS genes are encoded as part of a 16-gene operon, which includes may genes that putatively encode proteins involved in nitrogen metabolism, DNA repair, stress responses, and cell cycle processes. ChIP-seq analysis demonstrated that BvrR binds to more than 300 sites in the genome of B. abortus, and EMSAs confirmed direct binding to several of the identified regions. Further bioinformatic and biochemical (i.e., DNase footprinting) analyses defined a consensus DNA-binding sequence for BvrR, and the authors have developed a model for BvrR/BvrS-mediated control of metabolic systems (as well as other important virulence-related systems) in B. abortus.

Overall, the authors have performed a robust analysis of BvrR binding to DNA elements in B. abortus, and while many of the conclusions are supported by the data, there are some concerns about the data and conclusions. The specific concerns are:

-The authors have previously demonstrated that the BvrR/BvrS system is a transcriptional activator of virB. Here, the authors suggest that BvrR binds to the virB promoter at approximately -12 from the transcriptional start site. Mechanistically, this is difficult to understand. How does binding to that site promote transcription? In the same vein, in lines 558-568 the authors discuss the different potential binding sites for BvrR, but the data in Fig. 5 using limited DNA segments show that only one region is bound by BvrR. Is it possible that multiple binding sites are needed for optimal binding?

-Fig. 1 - This is a very minor point, but it is difficult to determine which gels/lanes correspond to the map and primer sets. Moreover, the authors have included a control to demonstrate that transcription stops that the 3' end of the message (i.e., the primer set represented by the black bar). Why is a similar control not included on the 5'-end of the message?

-Fig. 4 - Overall, the EMSAs are convincing, but there are some issues with some of them. For example, the binding to the virB promoter is highly variable between panels A and B. Why is this? It is understandable that differences exists between experiments, but in this case, the data are very difficult to interpret in terms of the competition controls when the control for those experiments looks nothing like the results in the panel A.

-Line 68 - Brucella replicates in a vacuole composed of (or associated with) the ER, and thus it may be incorrect to say that the bacteria replicate "within the ER."

Reviewer #2: This study aims to expand our knowledge of the genes controlled by the Brucella BvrR/S two component system.

While the experiments are well performed and the data presented solid, the conclusions are not fully supported by the data, more experimental work is needed.

Major Concerns

1 The authors write (L114) ‘We expand our knowledge of the BvrR/BvrS regulon, describing the genes controlled directly by this TCS and under conditions that mimic the intracellular environment confronted by B. abortus while trafficking to its replicative niche.’

This is not correct, the data show binding of Bvr-P to DNA, not regulation of gene expression. While there is some evidence that BvrR/S controls expression some genes, including virB and omp25, yo support this claim, it is essential that the authors provide data for the new set of genes that they claim to be controlled by BvrR/S.

2 The authors say that most BvrR-P binding sites are in regions upstream of the target genes. They also find binding sites in the virB4 and virB5 genes, several thousand bases into the operon. This is not at all discussed or commented on in the manuscript. How does this work? Are there internal promotors? This should be clarified.

3 Fig 3. Stress conditions increase binding of BvrR. Is this specific to BvrR. What happens with another TCS regulator, will it also bind to its targets more efficiently?

Other concerns

4 Fig 1 hard to follow with respect to text, there is a confusing mix of mix of gene names and gene numbers. It would be easier to follow if the figure showed the gene names referred to in the text. A more extensive color code could also help with calrity…so all pts genes in one color, unknown function in another etc.

5 The legend for Fig1B does not fit with the figure. First, the authors write that there are 31 primer pairs; this implied 31 PCR reactions, why are only 15 shown. If this is an English language problem and the authors meant 15 primer pairs (so 30 primers), where doses the number 31 come from?

6 The authors write ‘These co-transcription events happened at log and stationary growth phases…’ however they do not show data for different growth phases. What was the growth phase tested in Fig 1?

7 The introduction is rather confused.

Paragraph from L72-Are you talking about TCS in general or Brucella and BvrR/S? The refs suggest the latter the text the former.

L73 Define TCS

Paragraph from L80.

Here there are mixed references to TCS then PTS. It would be much clearer to introduce the Bvr family in the alphas and then talk about PTS.

8 L69 Bacteria then reach an autophagosome-like exit compartment where they are ready to egress from the host cell and start a new infection cycle [5]

Not an appropriate reference

9 L94 ‘Bacteria grown in a nutrient-rich medium at neutral pH (rich conditions), transiently activate BvrR through phosphorylation’

Modify to ‘When bacteria are grown in a nutrient-rich medium at neutral pH (rich conditions), BvrR is transiently activated through phosphorylation’

10 L392 typo rplI?

6. PLOS authors have the option to publish the peer review history of their article (what does this mean?). If published, this will include your full peer review and any attached files.

Reviewer #1: No

Reviewer #2: No

---

## [Author Response · Author response to Decision Letter 0]

28 Jun 2022

Dear Editor:

Enclosed you will find a modified version of the manuscript ID: PONE-D-22-09403: “The regulon of Brucella abortus two-component system BvrR/BvrS reveals the coordination of metabolic pathways required for intracellular life”, in a clean and tracked changes formats. 

We looked carefully into the reviewers’ comments and realized that more contextualization regarding some of the findings related to the non-canonical response regulators binding sites was need it. We have introduced such contextualization in the discussion section and for length’s sake we are referring the reader to literature related to this topic, which is certainly just being noticed in recent years. All comments were very helpful to improve the manuscript and considered. Figures were improved and figure captions modified accordingly. Additionally, a mistake in table 1 related to the bvrR locus tag was corrected, we improved Fig 5 legend and included minor drafting changes, all indicated in the track changes file. You will find a table answering each of the reviewers’ comments in the file named RESPONSE TO REVIEWER. 

Thank you for your consideration. Sincerely:

Caterina Guzmán Verri, PhD

Corresponding author

---

## [Decision Letter · Decision Letter 1]

13 Jul 2022

PONE-D-22-09403R1The regulon of *Brucella abortus* two-component system BvrR/BvrS reveals the coordination of metabolic pathways required for intracellular lifePLOS ONE

Dear Caterina,

Thanks for doing such a conscientious job with the revision!  Reviewer 2 has a couple of  minor suggestions and I think the one about mining the literature to see how many of  your BvrR targets have been identified in previous studies  is a good one. So, I am going to ask you to submit a revised manuscript that addresses this point. I agree with the reviewer that it will strengthen the paper.  Please submit your revised manuscript by Aug 27 2022 11:59PM If you will need more time than this to complete your revisions, please reply to this message or contact the journal office at plosone@plos.org. Please include the following items when submitting your revised manuscript:A rebuttal letter that responds to each point raised by the academic editor and reviewer(s). You should upload this letter as a separate file labeled 'Response to Reviewers'.A marked-up copy of your manuscript that highlights changes made to the original version. You should upload this as a separate file labeled 'Revised Manuscript with Track Changes'.An unmarked version of your revised paper without tracked changes. You should upload this as a separate file labeled 'Manuscript'.

Thanks for your patience with the review process. I look forward to seeing the revised manuscript!

Sincerely,

Marty Roop

Academic Editor

PLOS ONE

Journal Requirements:

Reviewers' comments:

Reviewer's Responses to Questions

**Comments to the Author**

1. If the authors have adequately addressed your comments raised in a previous round of review and you feel that this manuscript is now acceptable for publication, you may indicate that here to bypass the “Comments to the Author” section, enter your conflict of interest statement in the “Confidential to Editor” section, and submit your "Accept" recommendation.

Reviewer #1: All comments have been addressed

Reviewer #2: (No Response)

2. Is the manuscript technically sound, and do the data support the conclusions?

Reviewer #1: Yes

Reviewer #2: Yes

3. Has the statistical analysis been performed appropriately and rigorously? 

Reviewer #1: Yes

Reviewer #2: Yes

4. Have the authors made all data underlying the findings in their manuscript fully available?

Reviewer #1: Yes

Reviewer #2: Yes

5. Is the manuscript presented in an intelligible fashion and written in standard English?

Reviewer #1: Yes

Reviewer #2: Yes

6. Review Comments to the Author

Reviewer #1: Overall, the authors have appropriately addressed the comments raised in the previous review, and I do not have any addition major concerns.

Reviewer #2: The revised manuscript is greatly improved and much easier to follow.

The authors misunderstood my request to define TCS, I meant to explain what the abbreviation means (even though is is defined in the abstract). I would add this to line 73, where they first introduce BvrRS . The additional text they added, to describe what a TCS is, is very good!

While the authors argue that confirming the BrvRS targets experimentally can wait for another study, they should mine the available transcriptomic and proteomic studies comparing WT and TCS mutants to see whether any of their predicted targets have been found. This will strengthen the manuscript.

While the manuscript is understandable, the authors should have the text proof read by a native english speaker to correct the numerous small errors in grammar and syntax.

7. PLOS authors have the option to publish the peer review history of their article (what does this mean?). If published, this will include your full peer review and any attached files.

Reviewer #1: No

Reviewer #2: No

---

## [Author Response · Author response to Decision Letter 1]

24 Aug 2022

San José, August the 24th, 2022

Dear Editor:

Enclosed you will find a modified version of the manuscript ID: PONE-D-22-09403: “The regulon of Brucella abortus two-component system BvrR/BvrS reveals the coordination of metabolic pathways required for intracellular life”, in a clean and tracked changes formats. 

We would like to thank the reviewer and editor for the helpful comments to improve the manuscript. Below you will find a table answering each of the reviewer’s comments. There is a change in Fig 3, consisting of adding a Venn diagram for a better representation of the data mining result, as Fig 3C. The data supporting such representation is now an additional sheet in Table S4. 

Thank you for your consideration. Sincerely:

Caterina Guzmán Verri, PhD

Corresponding author

Responses to reviewer´s comments

Comment

Reviewer #2: The authors misunderstood my request to define TCS, I meant to explain what the abbreviation means (even though is is defined in the abstract). I would add this to line 73, where they first introduce BvrRS. The additional text they added, to describe what a TCS is, is very good! 

R/Thanks for the clarification. We defined the abbreviation as requested in line 75 of the tracked version

Comment

While the authors argue that confirming the BrvRS targets experimentally can wait for another study, they should mine the available transcriptomic and proteomic studies comparing WT and TCS mutants to see whether any of their predicted targets have been found. This will strengthen the manuscript. 

R/As requested, we performed data mining using the transcriptomic and proteomic studies with WT and TCS mutants. Detailed information is now presented as an additional excel sheet in table S4. Fig 3C was included and contains a Venn diagram presenting these results. Additionally, the following paragraphs were introduced in the results and discussion sections:

Line 379: “Fig 3C compares the results of this study and those reporting putative BvrR/BvrS targets, using transcriptomic and proteomic analysis of B. abortus 2308 and bvrR mutant strains. The three studies converged on identifying four common target genes, while our study compared only to proteomics or transcriptomics presented respectively 20 and 15 additional common target genes (S4 Table).”

Line 639 “The BvrR binding sites described in this work should be considered bonafide putative gene regulation sites. Some of these regions have been previously identified as putative BvrR/BvrS targets [15,16] and deserve further investigation”.

Comment

While the manuscript is understandable, the authors should have the text proof read by a native english speaker to correct the numerous small errors in grammar and syntax. 

R/We thank the reviewer for pointing this out. Indeed, there were numerous small errors in grammar and syntax. We have corrected as many as detected, as shown in the tracked version of the manuscript.

---

## [Decision Letter · Decision Letter 2]

28 Aug 2022

The regulon of *Brucella abortus* two-component system BvrR/BvrS reveals the coordination of metabolic pathways required for intracellular life

PONE-D-22-09403R2

Dear Caterina,

Thanks for making the requested modifications to your paper and I'm recommending approval. The paper will be formally accepted for publication once it meets any outstanding technical requirements. Please note that a reviewer suggested modifying a sentence for clarity, but I will leave that up to you whether on not you want to make that modification. 

Sincerely,

Marty Roop

Academic Editor

*PLOS ONE*

Additional Editor Comments (optional):

Reviewers' comments:

Reviewer's Responses to Questions

**Comments to the Author**

1. If the authors have adequately addressed your comments raised in a previous round of review and you feel that this manuscript is now acceptable for publication, you may indicate that here to bypass the “Comments to the Author” section, enter your conflict of interest statement in the “Confidential to Editor” section, and submit your "Accept" recommendation.

Reviewer #2: All comments have been addressed

2. Is the manuscript technically sound, and do the data support the conclusions?

Reviewer #2: Yes

3. Has the statistical analysis been performed appropriately and rigorously? 

Reviewer #2: Yes

4. Have the authors made all data underlying the findings in their manuscript fully available?

Reviewer #2: Yes

5. Is the manuscript presented in an intelligible fashion and written in standard English?

Reviewer #2: Yes

6. Review Comments to the Author

Reviewer #2: The manuscript is very much improved. I have one comment

The three studies converged on identifying four common target genes, while our study compared only to proteomics or transcriptomics presented respectively 20 and 15 additional common target genes (S4 Table).

I don't understand this sentence...there may just be punctuation problems (commas missing). Please revise.

Is this what you meant?

The three studies converged, identifying four common target genes, while our study, compared only to proteomics or transcriptomics, identified respectively 20 and 15 additional common target genes (S4 Table).

7. PLOS authors have the option to publish the peer review history of their article (what does this mean?). If published, this will include your full peer review and any attached files.

Reviewer #2: No

---

## [Editor Report · Acceptance letter]

8 Sep 2022

PONE-D-22-09403R2 

The regulon of *Brucella abortus* two-component system BvrR/BvrS reveals the coordination of metabolic pathways required for intracellular life 

Dear Dr. Guzmán-Verri:

I'm pleased to inform you that your manuscript has been deemed suitable for publication in PLOS ONE. Congratulations! Your manuscript is now with our production department. 

Kind regards, 

on behalf of

Dr. Roy Martin Roop II 

Academic Editor

PLOS ONE